# Relationship between Soluble (Pro)Renin Receptor and Renin Activity in Patients with Severe Heart Failure

**DOI:** 10.3390/jcm9124110

**Published:** 2020-12-19

**Authors:** Yoshifumi Ikeda, Kenta Tsutsui, Yoshihiro Yamada, Ritsushi Kato, Toshihiro Muramatsu, Takaaki Senbonmatsu

**Affiliations:** 1Department of Cardiology, Saitama Medical University International Medical Center, Saitama 350-1298, Japan; kent22e@gmail.com (K.T.); yyama@saitama-med.ac.jp (Y.Y.); ritsn@saitama-med.ac.jp (R.K.); toshi_m@saitama-med.ac.jp (T.M.); senbont@saitama-med.ac.jp (T.S.); 2Department, Research Administration Center, Saitama Medical University, Saitama 350-1298, Japan

**Keywords:** renin-angiotensin system, (pro)renin receptor, heart failure, diuretics

## Abstract

The (pro)renin receptor ((P)RR), which evokes renin activity with prorenin, is secreted extracellularly as soluble (P)RR (s(P)RR) and may participate in tissue renin-angiotensin system (RAS) activity in severe heart failure (HF) patients. The aim of this study was to determine whether s(P)RR is an adequate marker in severe HF patients treated with RAS inhibitors, beta-blockers, and tolvaptan. We enrolled 11 patients with severe HF between May 2013 and June 2014. First of all, furosemide of all patients was changed to tolvaptan with hydrochlorothiazide and then the treatment had been changed according to the patient’s condition. After 1, 3, 6, and 12 months, the variance of s(P)RR, plasma renin activity (PRA), plasma renin concentration (PRC), brain natriuretic peptide (BNP) and their association was investigated. Furosemide was restarted in five patients and two patients suffered cardiac death. PRA/PRC and s(P)RR were unchanged (PRA: 10.7 ± 13.9 to 12.8 ± 8.5 ng/mL/h; PRC: 347.1 ± 577.5 to 148.3 ± 123.8 pg/mL; s(P)RR: 28.2 ± 19.3 to 33.4 ± 22.4 ng/mL) and had no significant correlations (PRA and s(P)RR: *p* = 0.36; PRC and s(P)RR: *p* = 0.35). There was a significant positive correlation with a high correlation coefficient (CC) between PRA and PRC (*p* < 0.0001, CC = 0.76), and a negative correlation with weak CC between BNP and s(P)RR (*p* = 0.01, CC = −0.45). In conclusion, s(P)RR was always high and had no correlations with disease state and PRA/PRC in severe HF patients.

## 1. Introduction

Heart failure (HF) is a terminal condition in any heart disease, and the quality of life in HF patients gradually decreases. The five-year mortality of HF patients is almost 50% with optimal conventional medical therapies, such as lifesaving drugs, and many HF patients are hospitalized repeatedly. Reductions in renal blood flow, glomerular filtration rate (GFR), and sodium reabsorption by distal renal tubules through pump insufficiency induce sympathetic nerve activation in HF patients, leading to renin activation. Although this activation is a representative compensatory mechanism for the maintenance of life, chronic renin hyperactivity leads to the development of cardiovascular pathologic conditions by inducing proliferation of mesenchymal tissue [1,2]. Therapeutic agents such as renin-angiotensin system (RAS) inhibitors (angiotensin-converting enzyme inhibitors (ACEi), angiotensin II receptor blockers (ARB), or mineralocorticoid receptor antagonists (MRA)) and beta-blockers are effective for suppression of chronic renin hyperactivity in the treatment of patients with chronic HF.

Plasma renin activity (PRA) and plasma renin concentration (PRC) are standard indicators of renin activity, and reflect the circulating renin activity and concentration, respectively. Although several reports have demonstrated that PRA and PRC are predictors of prognosis in HF [3,4,5], these factors are strongly modified by RAS inhibitors and beta-blockers. There is also the concept of tissue renin activity, which acts as a limiting factor for tissue RAS activity. Although the detailed mechanisms for tissue renin activity are currently unknown, it is considered to be the rate-limiting step for RAS activity in organs. It is also believed that PRA and PRC may not reflect organ-level renin activity because circulating renin is only known to be secreted by juxtaglomerular cells in the kidney to date. Therefore, we have focused on the measurement of soluble (pro)renin receptor (s(P)RR) for evaluation of tissue renin activity in patients with chronic HF. (Pro)renin receptor ((P)RR) was discovered as a new member of the RAS family in 2002 [6].

Previous studies have reported that prorenin, a precursor of renin, bound with higher affinity than renin, suggesting that prorenin may also be a physiological endogenous ligand for (P)RR [7]. Binding of prorenin to (P)RR induces non-proteolytic renin activation of prorenin through a conformational change in its structure [8].

Interestingly, (P)RR is extracellularly secreted as s(P)RR. The 28-kDa soluble form is cleaved by the serine proteases furin and ADAM(a disintegrin and metalloprotease) 19 at the Golgi, leading to constitutive secretion of the extracellular form [9,10]. A previous study revealed that prorenin binds to s(P)RR and subsequently exerts renin activity through prorenin in plasma [11]. 

Despite a report that circulating s(P)RR levels may be unrelated to the levels of circulating renin or prorenin, a consensus has not been reached [12]. In severe HF patients, RAS of heart, kidney and vessel tissue is highly activated and in addition, they are under administration of drugs that exert a suppressive effect on renin activity. There is importance in investigating the changes in PRA/PRC and s(P)RR in HF patients under these complicated RAS conditions.

Meanwhile, diuretics appear effective at first glance [13]. However, there is no clinical evidence that any diuretics are effective for long-term use in chronic HF patients [13,14]. Sodium diuretics strongly stimulate renin activity through sodium loss and increased vascular resistance associated with fluid volume reduction. Thus, the development of a method for volume reduction that does not increase RAS activity is an important proposition in HF treatment. Tolvaptan, a selective Vasopressin V2 receptor blocker, belongs to the water diuretics without sodium loss [15].

Tolvaptan may be an ideal diuretic for HF from the viewpoint of RAS activity minimization [16]. However, single prescription of tolvaptan is not allowed, and therefore thiazide is selected as a combination partner rather than furosemide because thiazide is a weak sodium diuretic that shows only 5% GFR inhibition (i.e., 5% increasing effect on urinary volume).

The primary purpose of the present study was to investigate the relationship between s(P)RR, which may play an important role in tissue RAS activity, and common RAS parameters such as PRA and PRC, which are indicators of circulating RAS activity in severe HF patients. The secondary purpose was to investigate the correlations between therapeutic effects and kinetics of s(P)RR in HF patients treated with water diuretics.

## 2. Patients and Methods

This study was designed as a prospective cohort study (exploratory clinical study).

### 2.1. Patients

The inclusion criteria for the study were as follows: patients who provided written informed consent; patients aged > 20 years with moderate to severe symptoms exceeding NYHA (New York Heart Association classification) III and receiving standard medical therapies for HF (ACEi, ARB, beta-blocker, loop diuretic) in outpatient visits, and whose possibility of dose modification was estimated to be minimum; and patients hospitalized for acute exacerbation of chronic HF and expected to be discharged after HF treatment. Therefore, we judged that since the resistance to loop diuretics in all of the patients was high, the study had been started with a water diuretic that has the same diuretic effect as the loop diuretic and pharmacologically lower RAS activity than the loop diuretic according to the guidelines for acute decompensated heart failure. Finally, this protocol was approved by the Institutional Review Board of Saitama Medical University International Medical Center (12–148).

The exclusion criteria were as follows: patients considered to have a poor prognosis (estimated survival < 1 year); patients contraindicated for tolvaptan; patients with hypernatremia (>145 mEq/L serum Na); and patients with significant renal impairment (estimated GFR < 15 mL/min/1.73 m^2^; above stage V).

### 2.2. Methods

A scheme of the study protocol is shown in Figure 1. PRA, PRC, and s(P)RR were measured in serum after receiving patient consent. Blood samples were taken after a 30-min rest. The method for measurement of s(P)RR is described in Section 2.3. 

Patients taking furosemide 40 mg received tolvaptan 7.5 mg and hydrochlorothiazide 12.5 mg, while patients taking furosemide ≥ 60 mg received tolvaptan 15 mg and hydrochlorothiazide 12.5 mg. Serum PRA, PRC, and s(P)RR were measured at discharge and at 1, 3, 6, and 12 months after discharge. The clinical symptoms of the patients were evaluated by the NYHA classification. The NYHA classification was checked at hospitalization and discharge, and at 1, 3, 6, and 12 months during outpatient visits. Ultrasonic echocardiography was performed for evaluation of cardiac function at hospitalization, and at 6 and 12 months after discharge. The major clinical endpoint of cardiac death was defined as any cardiovascular death including left ventricular assist device (LVAS) implantation and HF hospitalization. The doses of tolvaptan and other diuretics were evaluated at hospitalization, and at 6 and 12 months after discharge. 

### 2.3. Method for Measurement of PRA, PRC and s(P)RR

Blood plasma was centrifuged at low temperature (4 °C), and the plasma was stored frozen.

Double-antibody radioimmunoassay was used for PRA measurement, and Immuno radio metric assay was used for PRC.

We measured s(P)RR using a soluble (Pro)renin Receptor Assay Kit (Immuno-Biological Laboratories Co. Fujioka, Japan). The kit contained a solid-phase sandwich ELISA(Enzyme Linked Immuno Solvent Assay) with two kinds of highly specific antibodies. Tetramethylbenzidine was used as a coloring agent/chromogen. 

The strength of the coloring was proportional to the quantity of s(P)RR. We created a concentration-absorbance curve using an analytical curve method and a spectrophotometer and calculated the s(P)RR values using an ARVO mx (Perkin Elmer, Waltham, MA, USA).

### 2.4. Statistical Analysis

The paired student *t*-test was chosen as the statistical test for changes in parameters over time, because this study was a prospective single arm test (there was no control group).

Continuous variables with a normal distribution are presented as mean ± standard deviation (SD), and continuous variables with a non-normal distribution are presented as mean (median (25th percentile–75th percentile)). The chi-square test was used for comparisons of categorical variables. 

Box plots and scatter plots were created using SPSS version 21 software (IBM Corp., Armonk, NY, USA). The box plots were used for comparisons that did not follow a normal distribution. The vertical bars in the box plots indicated the range and the horizontal boundaries of the boxes represented the first and third quartiles. Correlation analyses between the parameters in the scatter plots were performed by simple regression analysis.

### 2.5. Ethical Approval

The study was approved as a prospective study by the Institutional Review Board of Saitama Medical University International Medical Center (12–148). The study was evaluated by the Safety and Efficacy Committee, which consisted of two academic members and three non-academic members, to determine the progress, safety, and efficacy, and to judge whether or not to continue the study.

## 3. Results

Patients’ characteristic was shown in Table 1. Eleven patients with HF were registered for 14 months from May 2013 to June 2014. Investigation in patient 2 and 7 were discontinued from this study because their denovo malignant tumors were found at six and eight months. The total number of blood samples that could be evaluated was 64. Namely, four blood samples each for patient 2 and patient 7 and six blood samples each for the other nine patients.

The mean age of the patients was 76.4 ± 10.8 years, eight patients were male, and three patients were female. Five patients had cardiomyopathy (one with dilated phase of hypertrophic cardiomyopathy; four with dilated cardiomyopathy), three patients had valvular heart disease, two patients had coronary artery disease, and one patient had an aged heart. All patients were prescribed furosemide > 40 mg (mean dose: 51.0 ± 13.8 mg; maximum dose: 80 mg). The initial dose of tolvaptan was 15 mg in six patients and 7.5 mg in five patients. Furosemide was discontinued in all patients, followed by initiation of hydrochlorothiazide 12.5 mg as a combination drug for tolvaptan in all patients. Among the patients taking tolvaptan 7.5 mg, four patients increased to 15 mg during the observation period. Furthermore, five of eight patients whose final tolvaptan dose was 15 mg restarted furosemide for aggravation of body fluid retention and three patients were re-admitted at the end of the study.

Mean serum creatinine changed from 1.2 ± 0.3 to 1.3 ± 0.3 mg/dL (*p* = 0.07) and mean estimated GFR changed from 46.1± 15.8 to 38.0 ± 19.0 mL/min/1.73 m^2^ (*p* = 0.08). One patient received implantation of an LVAS and one patient died because of exacerbation of HF. The renal dysfunction in the latter patient was highly advanced, which was likely to be an important factor in the exacerbation of HF.

Figure 2 shows the changes in NYHA classification for all patients. The numbers in the bar graph show the numbers of patients categorized by NYHA classification. The number of NYHA II patients increased from zero to three, and the number of NYHA III patients decreased from five to three. Two of the patients who did not improve from NYHA IV suffered cardiac death. The mean NYHA classification showed improvement from 3.4 ± 0.5 to almost NYHA III (discharge: 3.0 ± 0.7, *p* = 0.006, 1 month: 3.1 ± 0.9, *p* = 0.10; 3 months: 3.1 ± 0.9, *p* = 0.10) in the first three months and finally changed to 2.8 ± 0.7. Although apparently significant at first glance, the two cardiac deaths were not included and it was thus judged that there was no significant difference. 

Meanwhile, mean left ventricular ejection fraction (LVEF) changed from 37.9 ± 20.4% to 34.9% ± 22.4% (*p* = 0.93), with no significant difference (Figure 3).

The previously administered RAS inhibitors and beta-blockers and the medication changes are listed in Table 2. Dose of Olmesartan was increased from 10 to 20 mg in patient 1, and carvedilol was increased from 0.625 to 1.25 mg in patient 3 due to hypertension. Carvedilol was discontinued in patient 5, while enalapril, carvedilol, and spironolactone were reduced in patient 6. In patient 8, enalapril was reduced from 5 to 2.5 mg, and spironolactone was stopped. The main cause of drug reductions and discontinuations in these patients was hypotension and its associated symptoms.

Figure 4 shows the kinetics for mean PRA, PRC, s(P)RR, and BNP as box plots. Mean initial PRA was 12.4 ± 15.0 ng/mL/h. Mean PRA after one month changed to 10.7 ± 13.9 ng/mL/h (*p* = 0.84), with no significant change thereafter (12 months: 12.8 ± 8.5 ng/mL/h, *p* = 0.83). PRC had large individual differences and was displayed as logarithm values. Mean initial PRC was 347.1 ± 577.5 pg/mL. Mean PRC after one month changed to 140.4 ± 212.5 pg/mL (*p* = 0.15), with no significant change observed during the study period (12 months: 148.3 ± 123.8 pg/mL, *p* = 0.40). Mean initial s(P)RR was 28.2 ± 19.3 ng/mL, which was obviously higher than the s(P)RR levels in previous studies [17,18,19,20]. Mean s(P)RR decreased to 20.2 ± 17.7 ng/mL at six months after tolvaptan therapy with no significant difference (*p* = 0.30), but increased again to 33.4 ± 22.4 ng/mL at 12 months (*p* = 0.97). There was no significant difference during the study period despite intensive therapy. Mean initial BNP was 406.5 ± 309.2 pg/mL. BNP was maintained at around 500 pg/mL throughout the study period with no significant changes (1 month: 570.9 ± 334.2 pg/mL, *p* = 0.16; 12 months: 494.4 ± 629.2 pg/mL, *p* = 0.38). The scatter plots for the correlations between all values of PRA and PRC, PRA and s(P)RR, PRC and s(P)RR, and BNP and s(P)RR for the whole study period are shown in Figure 5. There was a significant positive correlation with a high correlation coefficient (CC) between PRA and PRC (*p* < 0.001, CC = 0.76), but no significant correlations between PRA and s(P)RR (*p* = 0.36), PRA and BNP(*p* = 0.58), PRC and s(P)RR (*p* = 0.13), and There was a significant negative correlation between BNP and s(P)RR (*p* = 0.01), but the CC was not high(CC = −0.45).

Figure 6 shows the changes in PRA, PRC and s(P)RR, focusing on gender and age. Case 5 was described as a red solid line for a female over 80 years old. Other lines for over 80-year-olds were described as black and, for other females, were described as an orange line.

The continuous changes in PRA and PRC were very similar, in age and sex also, but s(P)RR had no particular tendency.

## 4. Discussion

### 4.1. Major Findings

The major findings of the present study were as follows. First, treatment with RAS inhibitors and beta-blockers was modified depending on individual patient situations, we performed a correlation analysis of each parameter by simple regression analysis. The mean basal level of s(P)RR in severe HF patients was very high and had no correlations with the levels of PRA and PRC during the entire study period. Meanwhile, PRA and PRC showed a significantly strong positive correlation, and s(P)RR and BNP showed a weak but significant negative correlation.

Second, the combination therapy of tolvaptan and thiazide tended to improve the NYHA classification in the first three months (mean NYHA: 3.5 ± 0.5 to 3.1 ± 0.9). Although two cardiac deaths were subsequently confirmed, it was finally judged that there was no treatment effect. The combination therapy did not improve LVEF and did not alter the s(P)RR level and RAS activity in patients with severe HF.

Third, the renal function of patients tended to worsen during the study period, albeit with no significant difference. One patient with exacerbation of congestive HF died during the study period and one patient received LVAS implantation.

### 4.2. Regulation of RAS Activity and s(P)RR in Severe HF

HF is a typical age-related disease, and its prognosis is extremely poor. Therefore, initial treatment is an important issue in a super-aging society. Chronic HF, particularly the classical reduced LVEF, develops after cardiac pathologic events such as myocardial infarction, myocarditis, or cardiomyopathy that reduce cardiac pump function. Under these situations, compensatory mechanisms such as activation of the sympathetic nervous system and RAS, which maintain cardiac output through increased retention of salt and water, peripheral arterial vasoconstriction, and increased contractility and inflammatory mediators, are involved in cardiac repair and remodeling. However, in the chronic HF state, sustained compensatory activation of neurohormones progresses the disease through deleterious effects on the heart and circulation. Therefore, development of therapy for HF can be said to have developed historically through suppression of the sympathetic nervous system and RAS activation.

ACEi, beta-blockers, and MRA are described as lifesaving therapies in some guidelines for drug management of chronic HF based on various lines of evidence in previous studies [21,22]. PRA and PRC are the standard indicators of renin activity and reflect circulating renin activity rather than tissue renin activity. Although several studies have identified PRA and PRC as predictors of prognosis for HF [3,4,5], RAS inhibitors, beta-blockers, and diuretics strongly modify their levels. Therefore, they are considered unreliable indicators under treatment states.

The RAS has multiple regulatory pathways, such as age, sex, retention of salt and water, peripheral arterial vasoconstriction, and increased contractility and inflammatory mediators, the results may be pharmacologically unexplainable.

In addition, RAS activity changes with malignancy, age and gender [23,24]. Surprisingly, the present study showed that there was a positive correlation between PRA and PRC, despite different patient backgrounds such as age and gender, disease progression, cancer development, and RAS inhibitor changes. In contrast, there was no correlation between s(P)RR and PRA/PRC.

We previously showed that s(P)RR induced non-proteolytic renin activation of prorenin extracellularly [9]. Although our previous study indicated that the circulating s(P)RR levels were unrelated to the levels of circulating renin or prorenin, the relationship between circulating s(P)RR levels and RAS activity remained unclear. Several clinical studies demonstrated that serum s(P)RR was related to the progression of diabetes and kidney disease [17,18]. Morimoto et al. [19] found that the mean s(P)RR level in essential hypertension patients was 19.0 ± 4.9 ng/mL, while another study showed that PRA/PRC gradually increased in pregnant women from early pregnancy to late pregnancy [20]. Specifically, s(P)RR increased from 10.7 to 29.7 ng/mL from early to late pregnancy. Furthermore, body fluid volume in pregnant women increased by about 1.5 times at late pregnancy, and blood pressure was also elevated. The authors indicated that s(P)RR may participate in increased fluid retention and hypertension. In recent years, there have been three important reports targeting patients with heart failure. Gong et al. report that the left ventricular mass index is an independent predictor of s(P)RR, and Obradovic et al. report a correlation between left ventricular remodeling and s(P)RR in elderly patients with heart failure. In addition, Fukusima et al. reported that s(P)RR was higher in heart failure patients than in healthy subjects, and that there was a positive correlation with BNP and a negative correlation with eGFR [25,26,27]. Generally, as HF becomes worse, body fluid volume increases and renal function decreases due to cardiac dysfunction. The simple hypothesis that PRA/PRC and s(P)RR are associated with fluid retention and renal function can suggest that worsening of HF is correlated with PRA/PRC and s(P)RR.

However, the results of this study did not support this simple hypothesis. Although PRA/PRC displayed the strong correlation, each of them did not show a correlation with BNP. The s(P)RR level was not significantly associated with PRA or PRC levels. Conversely, a significant negative correlation, but low correlation coefficient, was also observed between s(P)RR and BNP. Taken together, s(P)RR did not appear to be a marker that was able to assess the pathology of heart failure. All patients had not been treated with ARNI, which is an angiotensin receptor and neprylisin inhibitor and treatment-induced increase in BNP is unlikely. Therefore, BNP is the most reliable marker of heart failure in the present study [28].

Therefore, it seems that (P)RR and s(P)RR may be independent from RAS activity. For more than a decade, RAS-related (P)RR research including s(P)RR has been in chaos. Many promising initial articles were unable to be confirmed. Our results may be consistent with this chaos. A recent (P)RR-related article demonstrated that (P)RR contributed to integrity of vacuolar-type ATPase (V-ATPase), a multisubunit protein that regulates systemic acid-base homeostasis in vivo [29]. Moreover, Wnt(The name Wnt is a portmanteau coined from the names Wingless and Int-1)/β-catenin signaling was critically dependent on acidification of signaling endosomes by V-ATPase, suggesting that an indispensable role of (P)RR had reached a consensus. Meanwhile, Takahashi et al. [30] reported that (P)RR was widely expressed in the human brain. In the pituitary and frontal lobe, (P)RR was highly expressed and colocalized with vasopressin and oxytocin. Thus, (P)RR may regulate the central control of water–electrolyte metabolism and blood pressure. However, the brain is the organ with the lowest prorenin levels in the body. Taken together, s(P)RR may contribute to the cardiovascular pathologic state via the neurogenic system with V-ATPase and Wnt/β-catenin.

However, no loop diuretics have been shown to improve HF prognosis, because strong sodium diuresis leads to high RAS activity.

It was reported that tolvaptan had a RAS-regulating effect in two clinical trials, but tolvaptan was used combined with furosemide in these trials [31,32].

Combination therapies of tolvaptan and thiazide diuretics that can minimize Na excretion were considered more optimal strategies. Unfortunately, such a combination therapy could not maintain body fluid balance in severe HF patients and did not contribute to reductions in re-hospitalization and mortality in the present study. Furthermore, the combination therapy did not suppress PRA and PRC or s(P)RR in this study. The combination therapy failed to maintain control of fluid retention and required either tolvaptan increment or furosemide re-administration in almost half of the patients. Kinugawa et al. [33] performed a meta-analysis of clinical trials on tolvaptan and reported that tolvaptan did not impair renal function but reduced volume-dependent weight in patients. However, tolvaptan was used in combination with loop diuretics in almost all trials and this may be a combined effect with loop diuretics. The therapy was unable to sufficiently suppress RAS activity, and it is presumed that HF cannot be controlled in severe HF patients over NYHA III like the population in the present study.

### 4.3. Clinical Implications

s(P)RR was persistently elevated in severe HF patients and was independent of fluctuations in PRA and PRC levels. Thus, s(P)RR is unlikely to be affected by renin activity. Combination therapy with tolvaptan and thiazide for fluid control in severe HF patients over NYHA III is not an alternative method to loop diuretics.

### 4.4. Study Limitations

The present study has several limitations. The study was a single-center prospective study involving a small sample size and a short follow-up period. The mean NYHA above 3.4 was very high, the underlying etiology was varied, and the study was based on a very severe and limited patient population. The study was designed as a single-arm study, and there was no negative control.

Although this study has a very small number of cases, this study adopts paired statistical tests.

For a paired statistical test, if you specify a significance level of 5%, a static effect of 1.0, and a power of 80, the calculated minimum sample size is 10. Therefore, we believe this study meets the minimum criteria for statistical processing. However, the power of 80 is a limiting value and it is generally difficult to estimate a static effect for new parameters such as s(P)RR. The small number of cases is the biggest limitation of this study.

In the present study, s(P)RR did not correlate with PRA/PRC. Actually, we could not rule out the effects of the change in patients’ medication regimens, including tolvaptan treatment, which blocks the V_2_ receptor resulting in water diuretics; in that respect, it might have a negative effect on s(P)RR kinetics. This point has not been verified in this study and is an important study limitation. Theoretically water diuretics have a lesser effect on the RAS activity than that of Na diuretics, and we focused exactly on this point in this study. Despite changing the medication regimen, which indicates furosemide had been restarted in about half of the cases, there was no correlation between systemic PRA/PRC and s(P)RR; on the other hand, PRA and PRC showed a strong positive correlation in the linear analysis. Taken together, we determined that it might be reasonable to assume that changes of the medication regimen did not affect our outcome.

## 5. Conclusions

s(P)RR was very high and unaffected by PRA and PRC in severe HF patients over NYHA III. The regulators of s(P)RR are unknown and warrant further investigation.

## Figures and Tables

**Figure 1 jcm-09-04110-f001:**
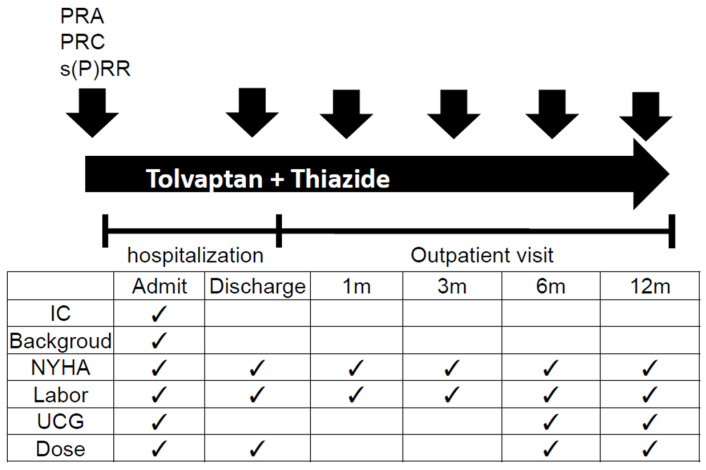
Study protocol. ✓ indicates informed consent, examinations and investigation of medication dose at timepoints. PRA: plasma renin activity; PRC: plasma renin concentration; s(P)RR: soluble (pro)renin receptor;

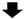
: measurement of RAS activity; Admit: hospital admission; IC: informed consent; NYHA: New York Heart Association classification; UCG: ultrasonic cardiography; Dose: dose of tolvaptan and other diuretics; m: month(s).

**Figure 2 jcm-09-04110-f002:**
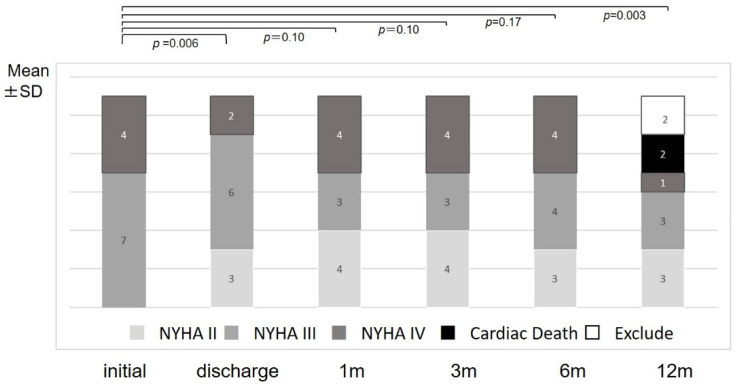
Numbers of patients categorized by NYHA classifications and mean NYHA classifications at initial visit, discharge, and 1, 3, 6, and 12 months after discharge. The numbers in the bar graph show the numbers of patients categorized by each NYHA classification. NYHA: New York Heart Association Classification; SD: standard deviation; m: month(s).

**Figure 3 jcm-09-04110-f003:**
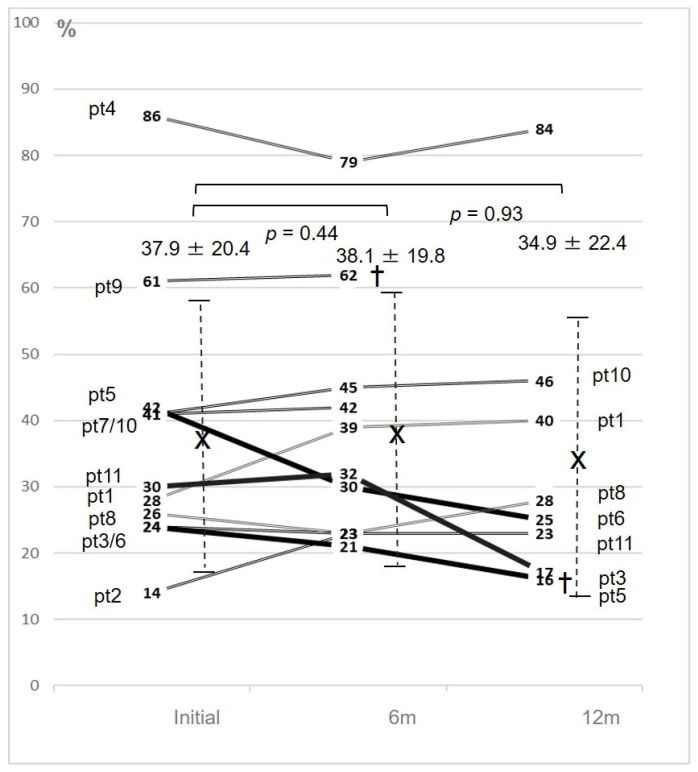
Changes in left ventricular ejection fraction (LVEF) for all patients and mean LVEF values during the study. The vertical axis shows LVEF (0-100%), and the horizontal axis shows time (month).The crosses indicate the mean values and the vertical dotted lines indicate the standard deviations. Three patients with marked LVEF changes are shown as black thick lines and characters, while the other patients are shown as grey double lines and characters. pt: patient; X: the mean value; †: cardiac death including left ventricular assist system (LVAS) implantation.

**Figure 4 jcm-09-04110-f004:**
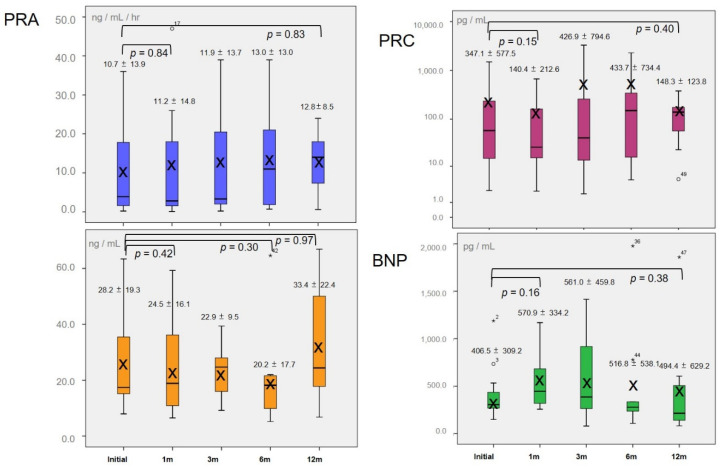
Changes in mean plasma renin activity (PRA), plasma renin concentration (PRC), soluble (pro)renin receptor (s(P)RR) and brain natriuretic peptide (BNP) levels shown as box plots. The time-dependent changes in mean PRA (blue), PRC (purple), s(P)RR (orange), and BNP (green) are shown. The horizontal lines in the box plots show the median values and the X show the mean values. The vertical axis for PRC is displayed as logarithm values.

**Figure 5 jcm-09-04110-f005:**
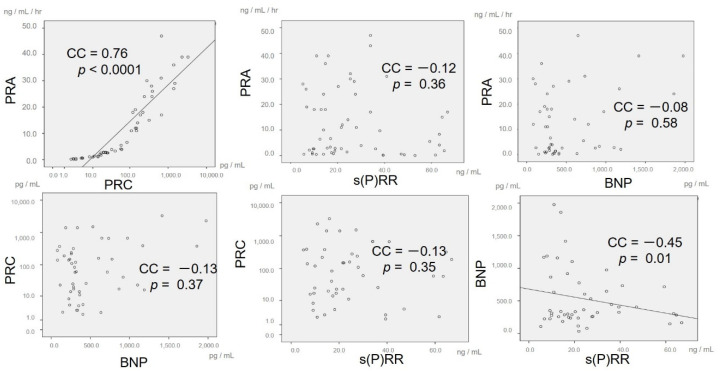
Scatter plots showing the correlations between all values of plasma renin activity (PRA) and plasma renin concentration (PRC), PRA and soluble (pro)renin receptor (s(P)RR), PRA and brain natriuretic peptide (BNP), PRC and BNP, PRC and s(P)RR, and BNP and s(P)RR during the entire study period.

**Figure 6 jcm-09-04110-f006:**
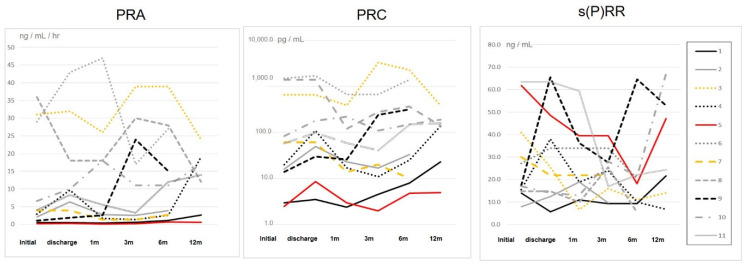
Changes in values of plasma renin activity (PRA), plasma renin concentration (PRC) and soluble (pro)renin receptor (s(P)RR) in each patient. The vertical axis of PRC was transcribed in a logarithm. Case 1,4 and 9 were described as black line for over 80 years old (case 1: solid line, case 4: dotted line, case 9: dashed line). Case 3 and 7 were described as orange line for female (case 3: dotted line, case 7: dashed line). Case 5 was described as a red solid line for a female over 80 years old. Case 5 as a gray dotted line, case 9 as a gray dashed line. Other cases were described as a gray line (case 2: solid line, case 6: dotted line, case 8: dashed line, case 10: dotted and dashed line, case 11: double line.

**Table 1 jcm-09-04110-t001:** Patient characteristics.

	Age	Sex	Etiology	Initial NYHA	Last NYHA	Initial EF (%)	Final EF (%)	Initial Cr/eGFR	BNP	PRA/ PRC	s(P)RR	Final Cr/eGFR	Frosemide Dose (mg)	Tolvaptan Dose (mg)	Final Tolvaptan	Furosemide Restart	Re- Admission	Outcome	
1	89	m	DCM	III	II	28	40	0.9/63.9	337	0.5/3.0	15	1.6/30.2	60	15	15	yes	no	Study Fin	
2	87	m	DCM	III	NA	14	NA	1.5/35.9	1188	2.1/16.0	8	NA	40	7.5	NA	NA	NA	Excluded	
3	67	f	dHCM	IV	IV	24	16	1.2/35.8	734	31.0/660	40.9	1.5/25.4	40	7.5	15	yes	yes	Study Fin	
4	86	m	Aged heart	III	III	86	84	1.2/44.3	287	2.9/20.0	15.5	1.5/34.3	60	15	15	yes	no	Study Fin	
5	81	f	VD	III	II	42	25	1.2/34.5	149	1.2/34.5	61.7	0.9/43.7	40	7.5	15	no	no	Study Fin	
6	58	m	DCM	IV	†(IV)	24	23	0.9/65.1	533	0.9/65.1	27.1	1.1/53.0	60	15	15	no	yes	LVAS imp	
7	72	f	VD	IV	NA	41	NA	1.4/30.0	203	1.4/30.0	30	NA	60	15	NA	no	NA	Excluded	
8	71	m	CAD	IV	III	26	28	0.9/68.0	186	0.9/68.0	14.8	0.75/77.9	40	7.5	15	no	yes	Study Fin	
9	88	m	VD	III	†(IV)	61	NA	1.6/31.7	250	1.6/31.7	17.4	2.7/17.5	40	7.5	7.5	yes	yes	Death by HF	
10	63	m	CAD	III	II	41	17	0.9/64.7	299	0.9/64.7	17.9	1.0/56.0	80	15	15	no	no	Study Fin	
Av	76.4 ± 10.8			3.5 ± 0.5	2.8 ± 0.8	37.9 ± 20.4	34.9 ± 22.4	1.2 ± 0.3/	426.8 ± 318	1.2 ± 0.31/	28.2 ± 20.1	1.3 ± 0.3/	51.0 ± 13.8	10.9 ± 3.9	13.3 ± 3.3				
46.1 ± 15.8	46.1 ± 15.0	42.0 ± 18.1	
						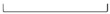	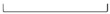										
						*	**										

The table shows the characteristics of the 11 patients. All values for mean and standard deviation were calculated including patients 2 and 7 until their de novo malignancies were found during the study (colored in dark gray). m: male; f: female; NYHA: New York Heart Association Classification; LVEF: left ventricular ejection fraction; BNP: brain natriuretic peptide; PRA: plasma renin activity; PRC: plasma renin concentration; s(P)RR: soluble (pro)renin receptor; Cr: creatinine; eGFR: estimated glomerular filtration rate; DCM: dilated cardiomyopathy; dHCM: dilated phase of hypertrophic cardiomyopathy; VD: valvular heart disease; CAD: coronary artery disease; NA: not applicable; LVAS imp: left ventricular assist system implantation; HF: heart failure; SD: standard deviation.* EF: *p* = 0.93; ** Cr: *p* = 0.07; eGFR: *p* = 0.08.

**Table 2 jcm-09-04110-t002:** Other medications and medication changes.

case	ACEI/ARB	Dose (mg)	BB	Dose (mg)	MRA	Dose (mg)	
1	Olmesartan	10 to 20	Carvedilol	5	NA		
2	NA		Carvedilol	5	NA		
3	Enalapril	2.5	Carvedilol	0.625 to 1.25	Spironolactone	50	
4	Losartan	25	Carvedilol	5	Eplerenone	25	
5	Enalapril	2.5	Carvedilol	1.25 to off	NA		
6	Enalapril	10 to 5	Carvedilol	7.5 to 5	Spironolactone	50 to 25	
7	Enalapril	5	NA		Spironolactone	50	
8	Enalapril	5 to 2.5	Bisoprolol	1.25	Spironolactone	25 to off	
9	Valsartan	40	Carvedilol	5	NA		
10	Enalapril	2.5	Carvedilol	3.75	NA		
11	NA		Carvedilol	5	Spironolactone	25	

The table shows other drugs related to renin-angiotensin system (RAS) activation and medication changes. e.g. “ 10 to 20 “ mean that medication dose hase been changed 10 mg to 20 mg during study period. ACEi: angiotensin-converting enzyme inhibitor; ARB: angiotensin receptor blocker; BB: beta-blocker; MRA: mineralocorticoid receptor antagonist; NA: not applicable.

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
