# Peer review of "Relationship between Soluble (Pro)Renin Receptor and Renin Activity in Patients with Severe Heart Failure"

_jcm, 2020, doi:10.3390/jcm9124110_

Round 1

Reviewer 1 Report

No further comments

Author Response

Thank you for your careful review of the manuscript "Relationship between soluble (pro) renin receptors and renin activity in patients with severe heart failure". And we are deeply grateful for accepting our paper to be published in journal clinical medicine.

Reviewer 2 Report

Authors Ikeda et al. are to be congratulated for a nicely presented study and manuscript. They investigate the effect of tolvaptan and thiazide administration on n=11 patients with NYHA III CHF and conclude that no correlation between sPRR and PRA and PRC existed. This finding refutes a previously held hypothesis that PRA/PRC and sPRR were associated with fluid retention and renal function leading to worsening CHF. Thus tolvaptan and thiazide are not a substitute for loop diuretics.

The manuscript is well written in English, and the figures are easy to interpret.

Some very minor suggestions and thoughts that the authors may want to consider:

  1. In the abstract, please define abbreviations "PRA" and "PRC".
  2. Is it possible that adjustments to the patients' medication regimens may impact the negative conclusions?
  3. Is there any thought why the thiazide and tolvaptan combination helped improved LVEF for the first 3 months only?

Congratulations again to the authors for a well written manuscript.

Author Response

#Response of Reviewer 2

Thank you for your careful and insightful review of our manuscript, we have taken each of your concerns seriously and have addressed each of them. Our response to each of your concerns is as follows:

#Response of concerning 1

Thank you for your advice. I really agree your suggestion.

Following your suggestions, we have modified the summary (page 1, line 18-19).

#Response of concerning 2

Your concern about the issue is really appropriate and important.

Actually, it cannot be ruled out effects of changing of the patients' medication regimens including tolvaptan treatment which blocks V2 receptor resulting in water diuretics, in that respect, it might have a negative effect on SPRR kinetics. This point has not been verified in this study, and there is no literature discussing the relationship between vasopressin secretion and SPRR, to the best of our knowledge.  Theoretically water diuretics have lesser effect on the RAS  activity than that of Na diuretics, and we exactly focused on this point in present study. Despite changing medication regimen that indicates furosemide had been restarted in about half of the cases, there was not the correlation between systemic PRA / PRC and sPRR, on the other hand, PRA and PRC showed strongly positive correlation in the linear analysis. Taken together, we determined that it might be reasonable to assume that changes of medication regimen did not affect our outcome. However, this is a study limitation, so we added it to the limitation on page 12-13, line 364-373.

#Response of concerning 3

Thank you for your meaningful comments. In fact, as an interim report of the study, we reported good results of this combination therapy after 3 months follow up in  European Society of Cardiology Heart Failure Congress in 2016 at Florence. The strong diuretic effect of tolvaptan may have eliminated congestion and improved systemic hemodynamics in the short term. However, we conclude that there is no sufficient effect in the medium to long term.

Reviewer 3 Report

The present manuscript, written by Yoshifumi Ikeda and colleagues, is an original paper on the difficult topic of the renin-angiotensin system (RAS) activity in severe heart failure (HF). The authors measured 12-month kinetics of less explored parameters of RAS system, namely (pro)renin receptor ((P)RR) and soluble (P)RR (s(P)RR) in 11 HF patients. The authors did not observe any  correlations between those new parameter and disease state and plasma renin activity (PRA) and plasma renin concentration (PRC).

The authors studied the complex relations within RAS system in HF. Uniquely, the parameters under study are far less known. It is not for the fact that overall the study results should be termed ‘negative’ and perhaps the whole hypothesis seems to be questionable, it is that even if the study results had been ‘positive’, there are still numerous concerns that make the interpretation of the results very difficult, if not impossible.   

Concerns:

  1. Lack of the control group is a major limitation, that basically makes this study uninterpretable. (Pro)renin receptor ((P)RR) and soluble (P)RR (s(P)RR) are not standard HF parameters and their normal concentrations and 12-month kinetics are barely known. In fact, the authors perhaps should even include two control groups: (1) healthy volunteers, and (2) cardiac patients with mild symptoms intensity, e.g. stable coronary artery disease, HF with preserved ejection fraction, etc. in order to present the “normal” concentrations and kinetics of those parameters.
  2. The authors explained their rationale for loop diuretics withdrawal and replacing with the vassopresin antagonist – Tolvaptan. Although there is a scientific background of such an usual approach, nonetheless, this again create huge confusions. How the study can be interpreted and more importantly translate into practice when patients were treated against the guidelines? Again, if the authors recruited two groups of HF patients – one treated according to the guidelines (i.e. with loop diuretics) and the second one – with Tolvaptan, this would be much more informative. Besides that, adherence to the guidelines is fundamental in such a severe HF cohort (and lack of proven diuretics may have also translated into their poor outcomes).

Author Response

#Response of concerning 1 and 2

Thank you for your careful review of our manuscript. Your concern about the issue is really appropriate and important. As your pointed out, tolvaptan is not listed as the first line of diuretics in the main guidelines as a diuretic for HF. However, tolvaptan is covered by Japanese medical insurance for the treatment of HF, and this study was designed in a single-arm study to demonstrate the expected adverse effects of tolvaptan on RAS activity without setting a control group, because of the ethical guidelines, it is extremely difficult to conduct an interventional clinical study.

Our answer to your two concerns is overlap. Therefore, we will write the reply together as follow.

Regarding the use of diuretics in the guidelines of your concern 2, it is true that the use of loop diuretics is listed as the first line of the main guidelines for HF therapy. However, it aims to improve symptoms by relieving congestion, and there is no evidence to improve the prognosis of life. Loop diuretics are not selected based on evidence, but because of their strong diuretic properties, high convenience with dose-dependency, and enough experience with long-term use. Rather, loop diuretics worsen the prognosis in dose-dependency. In the EVEREST study, tolvaptan did not improve the prognosis of HF compared to placebo but showed strong diuresis. Point of view from pharmacology, tolvaptan, same as loop diuretics, has a diuretic effect that suppresses 20% of GFR. A strong diuretic effect is an important factor in the selection of diuretics for HF therapy. Therefore, we determined that tolvaptan, which  theoretically has weaker RAS activity compared with that of loop diuretics, is well tolerated as an alternative to loop diuretics. Finally, this study was approved by the Institutional Review Board of our institute. In fact, tolvaptan can be used as a treatment for HF in Japanese insurance practice, and tolvaptan was described as treatment for HF for Class IIa recommendation in the current guideline of Japanese Circulation Society.

Next, we give a reply regarding your concern 1. As you pointed out, although there is no control group in the present study, we fully understand that intervention clinical study with control group has more statistical power reflect clinical outcomes. However, as mentioned above, the purpose of this study was to investigate the effect of avoiding RAS-activation of Na diuresis, which has a large effect on the treatment of HF, in severe heart failure with a poor prognosis. In addition, since the target was patients with severe heart failure, the patients' medication regimens had to be changed in response to the patient's condition. Therefore, we performed linear analysis without a control group, designed the study with a single arm, adopted the evaluation of PRA / PRC as a systemic RAS dynamics as an important endpoint, and s(P)RR, which may be a parameter of tissue RAS activity.

New sentence was added to the introduction (page 2, line 76-79).

Round 2

Reviewer 3 Report

This is a second round review of the manuscript, written by Yoshifumi Ikeda and colleagues, on the difficult topic of the renin-angiotensin system (RAS) activity in severe heart failure (HF). Based on the careful evaluation of the paper, the previous recommendation was to “reject” the paper. The comments send by the authors do not improve the manuscript in any way suggested by the reviewer. Thus, the previous recommendation is maintained.

Author Response

Thank you so much for great suggestion. However, since this is an observational study as mentioned in our manuscript, we cannot modify to fit an interventional study.

Also our study protocol does not deviate from the guidelines. Page 3, lines 90-93 have been modified to accommodate your indications.

In addition, we found three previous interventional studies about relationship between HF and s(P)RR that could be contrasted with our study, so we changed the references and discussion.

It is a limitation that our study is an observational study rather than interventional study, but we have thought that even this observational study is quite important as a source of contradiction with these studies.

Please confirm the sentence on page 11, line 306-311 and new reference 25, 26 and 27. We have changed the reference number accordingly on page 14-15.

This manuscript is a resubmission of an earlier submission. The following is a list of the peer review reports and author responses from that submission.

Round 1

Reviewer 1 Report

Ikeda and colleagues studied the Relationship between soluble (pro)renin receptor and 3  renin activity in patients with severe heart failure and they concluded that s(P)RR was always high and had no correlations with disease state and PRA/PRC in severe HF patients.

I have following major concern regarding the conclusion of study:

  1. Authors included only 9 NYHA III patients in the study, out of which 2 died during the study period so authors presented the data from remaining 7 patients. From clinical perspective, the sample size is very small to conclude that treatment regime has no effect on s(P)RR in relation to disease state and PRA.
  2. Authors should increase the sample size significantly to conclude the their study as clinically relevant.

Reviewer 2 Report

   The authors investigated if sPRR can be a pathological marker of heart failure by employing patients above NYHA III receiving RAS blockages, beta-blockers and tolvaptan. Although the results indicate that sPRR levels do not reflect the degrees of the disease in the patients, this study provides some valuable findings which will help further studies and facilitate establishment of biomarkers of the disease. However, there are still several concerns which must be appropriately addressed.

Comments

  1. The authors mention that sPRR can activate (pro)renin. Provide a firm rationale for focusing on sPRR as a marker of HF although PRA levels were not altered in the patients during the tested period (12 months). In addition, a published article which demonstrated activation of human (pro)renin by sPRR under in vivo conditions must be cited.
  2. Systemic and local RAS have exhibited sex differences. While this study includes only two female subjects, it would be interesting to evaluate sPRR levels in male and female separately. Likewise, comparisons of the changes in sPRR between younger age (e.g. <80 YO) and older age might be also interesting since incidence of HF and RAS regulation are age-dependent, respectively.
  3. Provide details of method for measurements of PRA and PRC.
  4. The Introduction and Discussion sections are a bit long. Thus, Introduction section can be summarized and shortened by omitting descriptions regarding general backgrounds. Moreover, many background information in the discussion section can be moved to the shortened Introduction section.